# On the use of reference mass spectra for reducing uncertainty in source apportionment of solid fuel burning in ambient organic aerosol

Chunshui Lin[1,2,3], Darius Ceburnis[1], Anna Trubetskaya[4], Wei Xu[1], William Smith[5], Stig Hellebust[6], John Wenger[6], Colin O'Dowd[1*], and Jurgita Ovadnevaite[1*]

[1]School of Physics, Ryan Institute's Centre for Climate and Air Pollution Studies, National University of Ireland Galway. University Road, Galway. H91 CF50, Ireland
[2]State Key Laboratory of Loess and Quaternary Geology and Key Laboratory of Aerosol Chemistry and Physics, Chinese Academy of Sciences, 710061, Xi'an, China
[3]Center for Excellence in Quaternary Science and Global Change, Institute of Earth Environment, Chinese Academy of Sciences, Xi'an 710061, China
[4]Department of Chemical Engineering, Aalto University, 02150 Espoo, Finland
[5]School of Electrical, Electronic and Mechanical Engineering, University College Dublin, D04V1W8 Dublin, Ireland
[6]School of Chemistry and Environmental Research Institute, University College Cork, T23XE10 Cork, Ireland

Correspondence to: Colin O'Dowd (colin.odowd@nuigalway.ie) and Jurgita Ovadnevaite (jurgita.ovadnevaite@nuigalway.ie)

**Abstract.** Reference mass spectra are routinely used to facilitate source apportionment of ambient organic aerosol (OA) measured by aerosol mass spectrometers. However. source apportionment of solid fuel burning emissions can be complicated by the use of different fuels, stoves, and burning conditions. In this study, the organic aerosol mass spectra produced from burning a range of solid fuels in several heating stoves have been compared using an aerosol chemical speciation monitor (ACSM). The same samples of biomass briquettes and smokeless coal were burnt in a conventional and Ecodesign stove, while different batches of wood, peat, and smoky coal were also burnt in the conventional stove and the OA mass spectra compared to those previously obtained using a boiler stove. The results show that although certain ions (e.g., $m/z$ 60) remain important markers for solid fuel burning, the peak intensities obtained at specific $m/z$ values in the normalized mass spectra were not constant with variations ranging from <5% to >100%. Using the OA mass spectra of peat, wood, and coal as anchoring profiles and the variation of individual $m/z$ values for the upper/lower limits (the limits approach) in the Positive Matrix Factorization (PMF) analysis with the Multilinear Engine algorithm (ME-2), the respective contributions of these fuels to ambient sub-micron aerosols during a winter period in Dublin were evaluated and compared with the conventional $a$ value approach. The ME-2 solution was stable for the limits approach with uncertainties in the range of 2-7%, while relatively large uncertainties (8-29%) were found for the $a$ value approach. Nevertheless, both approaches showed good agreement overall, with the burning of peat (39% vs 41%) and wood (14% vs 11%) accounting for the majority of ambient organic aerosol during polluted evenings, despite their small uses compared to electricity and gas. This study, thus, accounts for the source variability in ME-2 modelling and provides better constraints on the primary factor contributions to the ambient organic aerosol estimations. The finding from this study has significant implications for public health and policymakers considering that it is often the case that different batches of solid fuels are often burned in different stoves in real-world applications.

## 1 Introduction

Aerosol particles adversely affect human health and play an important role in the climate system (Fuzzi et al., 2015; Hallquist et al., 2009; Zhang et al., 2015). A better understanding of their sources is crucial to develop cost-effective air quality control strategies, as well as to better constrain their corresponding climate effects (An et al., 2019; Shrivastava et al., 2017). Aerosols can be broadly categorized into primary aerosols, which are directly emitted from sources such as biomass and fossil-fuel burning, and secondary aerosols, which are formed in the atmosphere from precursor gases, such as volatile organic compounds, ammonia, sulfur, and nitrogen dioxide. Organic aerosol (OA) is a major component of ambient particulate levels in the atmosphere and the aerosol chemical speciation monitor (ACSM) is regularly used to quantitatively evaluate the contribution of its various primary and secondary sources. This approach to OA source apportionment uses receptor models such as positive matrix factorization (PMF) with the multilinear engine algorithm (ME-2) (Canonaco et al., 2013; Canonaco et al., 2015; Paatero, 1997, 1999). However, the selection of reference OA mass spectra or profiles in the ME-2 modelling can be a significant source of uncertainty (Canonaco et al., 2013; Lanz et al., 2008). Using reference profiles that are representative of specific local sources can reduce the uncertainty of source apportionment (Lin et al., 2017), while the use of more generic profiles from the literature can sometimes cause substantial uncertainty (Hopke, 2016). However, even for local sources, the profiles of the emissions may vary significantly e.g., for biomass burning due to the use of different fuels, stoves, and burning conditions, causing uncertainty in the ME-2 based source apportionment.

Residential solid fuel burning, such as biomass burning and coal combustion, has been reported to be an important source of particulate pollution, affecting local and regional air quality in both developing and developed countries across the world (Crippa et al., 2014; Li et al., 2017). Dublin is a moderately sized city in western Europe with a population of around 1 million. Recent studies in Dublin show that residential burning of solid fuels – mainly peat and wood, but also coal to a lesser degree – is a significant source of ambient organic aerosol (OA) during the heating season (Lin et al., 2019; Lin et al., 2018). In a case study, Lin et al. (2018) show residential heating and particularly peat and wood burning caused an extraordinarily high concentration (over 300 μg m$^{-3}$) of submicron aerosol, affecting air quality on a local to regional scale in suburban Dublin. Source attribution of the measured OA to different types of solid fuels was performed using reference profiles from locally sourced fuels (Lin et al., 2017) as the anchoring profiles in the ME-2 modelling (i.e., the *a* value approach (Canonaco et al., 2013)) The reference profiles for solid fuels was obtained from a combustion experiment using a boiler stove with no emission control (Lin et al., 2017). However, the question remains on how these reference profiles vary with stove type and what uncertainties this variation causes in the ME-2 modelling.

In this study, mass spectral signatures of OA emissions from combustion of the same batch of wood, peat, smoky coal, biomass briquettes, and smokeless coal in two different heating stoves - a conventional stove and an Ecodesign stove (Trubetskaya et al., 2021) - were characterized using an ACSM. The corresponding implications for ambient OA source apportionment are discussed. Moreover, through comparison with a different batch of wood, peat, and smoky coal combusted in a boiler stove (Lin et al., 2017), the variation of the source profiles for these solid fuels was further characterized. The

obtained source profiles were subsequently used as upper/lower limits (i.e., the limits approach) in the ME-2 modelling of ambient OA in Dublin from 1 November 2016 to 31 January 2017, and compared with the *a* value approach (Canonaco et al., 2013). This enabled determination of the contribution of peat, wood, and coal burning to ambient OA, as well as the corresponding uncertainties.

## 2. Materials and Method

### 2.1 Combustion experiments

Two different testing stoves - a conventional stove and an Ecodesign stove - were used for the burning experiments in this study (Schematic S1). The description of the stoves and experimental setup are detailed in Trubetskaya et al. (2021). Briefly, the conventional stove uses a primary air supply through an inlet below the door of the stove, while the Ecodesign stove draws both primary and secondary air through two valves on the rear side of the stove. Five fuel types were tested including wood, peat, smoky coal, biomass briquettes, and smokeless coal (Table 1). Specifically, wood logs were cut from softwood grown in Ireland; peat was obtained from the peatland in Leitrim, Ireland, and was naturally dried before testing; smoky coal (Silesia, Poland) were purchased from local retail outlets (Trubetskaya et al., 2021); biomass briquettes and smokeless coal (i.e., Ecobrite ovoids) were manufactured at Arigna Fuels (Carrick on Shannon, Ireland). For each burning experiment, 3.5 kg of the test fuel was placed in the stove and 100 g firelighters (TESCO, Ireland) were used to ignite the solid fuels. In order to avoid sampling of aerosol emissions from firelighter burning, the ACSM measurements were not started until the firelighters were burned out (15 min after ignition). The combustion experiment lasted 1-3 hours depending on the fuel and stove types. The time resolution of ACSM was set to 2 min to capture the variation of the combustion emission. The stove was cleared of residue following the combustion of each fuel. The particle samples generated from the combustion of fuels were extracted from a port in the chimney, 112 cm above the stove. The sampling line was made of ordinary ½ inch copper pipe, with a total length of 2 m. After drawing the flue gas through a $PM_{2.5}$ cyclone and moisture trap, a diluter (DI-1000; Dekati Ltd) was used. Through the diluter, the raw flue gas was diluted with compressed clean cool air with a dilution range of 70-200:1. The cooled, diluted sample was then split and fed into PM sampling system described below.

### 2.2 Instruments

A quadrupole ACSM (Aerodyne Research Inc.) (Ng et al., 2011b) was used to characterize the mass spectral signatures of organic aerosol particles produced from solid fuel burning. The operation principles of the ACSM are detailed in Ng et al. (2011b). In this study, a $PM_{2.5}$ cyclone was deployed to remove coarse particles. The aerosol particles were passed through a Nafion dryer (Perma Pure PD-50T-24SS) before they entered the ACSM. A $PM_1$ aerodynamic lens was used to focus the submicron particles into a narrow beam. In the vacuum chamber of the ACSM, the particle beam was deposited on the heated surface (600 °C) where the non-refractory materials including OA, sulfate, nitrate, ammonium, and chloride were vaporized. Note that chloride in the aerosol emission from biomass burning is often present as KCl which vaporizes slowly at 600 °C,

requiring a non-standard treatment of the ACSM chloride data (Lee et al., 2010). In this study, we focused on the mass spectral

profiles of OA emissions, and the slow vaporization issue was not accounted for (Lee et al., 2010). Upon deposition at 600 °C, the resulting vapor for the non-refractory species was ionized by electron impact (70 eV) and the gaseous ions were analyzed using the quadrupole mass spectrometer. ACSM was calibrated following the procedure described by Ng et al. (2011). Briefly, a Scanning Mobility Particle Sizer (SMPS, TSI 3938) was used to size-select (300 nm) the atomized ammonium nitrate or ammonium sulfate, which was subsequently fed into ACSM system. For the fingerprinting experiments, the OA mass spectra

from each testing stove were averaged, representative of the fingerprints of the different types of solid fuel burning in different stoves.

For ambient measurement of submicron aerosol ($PM_1$) in Dublin, an ACSM and Aethalometer (AE33, Magee Scientific, (Drinovec et al., 2015)) were deployed at University College Dublin (UCD) from 1 November 2016 to 31 January 2016 (Lin et al., 2018). The ACSM sampling site is ~5 km south to the Dublin city center and is ~500 m away from the nearby road (Fig.

S1). ACSM measurements were conducted on the roof of the O'Brien Centre for Science building (~ 30 m above the ground). Previous studies conducted at the same sampling site show the aerosol population was mainly affected by the heating emissions but with a relatively minor contribution from traffic or cooking emissions (Lin et al., 2020; Lin et al., 2018). The aethalometer measured the light absorption of the particles collected on a filter at seven wavelengths (370, 470, 520, 590, 660, 880, and 950 nm). The mass absorption cross section of 7.77 $m^2$ $g^{-1}$ was used to calculate the BC mass concentration based on the changes

in optical attenuation at 880 nm (Drinovec et al., 2015). Local $PM_{2.5}$ measurements were obtained from EPA Ireland who operates an air quality monitoring station in Rathmines (www.airquaity.ie; Last access: 1 September 2021), ~3 km west of the ACSM sampling site. Meteorological parameters were from the Meteorological station at Dublin airport.

## 2.3 OA source apportionment

Positive matrix factorization (PMF; (Paatero, 1997)) with the multilinear-engine (ME-2; (Paatero, 1999)) on the interface of

SoFi (version 6.F1) (Canonaco et al., 2013) was employed to apportion the measured OA into different factors by constraining their corresponding reference profiles. The PMF model in matrix notation is defined as:

$$X = GF+E,$$

where the measured matrix X is approximated by the product of G and F, while E is the model residual. The PMF output is a set of factors representing factor profiles (mass spectra) and their corresponding time series. For unconstrained or free PMF,

no priori information about the source profiles is required to obtain a mathematical solution. However, the PMF solutions are not mathematically unique due to rotational ambiguity. Instead, interpretation of the factors (e.g., source type and contribution) is usually carried out with reference to known profiles of source emissions or typical diurnal patterns (Ulbrich et al., 2009). Nevertheless, the unconstrained PMF can experience difficulties in separating aerosol sources with temporal covariations, resulting in unrealistic or highly mixed factors (Canonaco et al., 2013).

As shown in Fig. S2, unconstrained or free PMF suffered from factor mixing due to temporal covariation of the candidate factors (i.e., all increasing in the evening corresponding to the time of domestic heating activities). To evaluate the contribution

of different types of solid fuels, source profiles obtained from the combustion experiments can be used as the anchoring factor-profiles (i.e., reference mass spectra) in the ME-2 algorithm (Canonaco et al., 2013). However, without extensive and objective analysis, both free PMF and ME-2 analysis can fail to apportion the sources accurately especially when the reference mass spectra can be complicated by the use of different fuels, stoves, and burning conditions. The "*a* value" approach (Canonaco et al., 2013) allows a certain degree of variation from the anchoring profiles. For example, an *a* value of 0.3 corresponds to 30% variation, while an *a* value of 1 is equivalent to the completely unconstrained (or free) PMF situation. In the conventional *a* value approach, the same *a* value is applied to all of the *m/z* values at the same time. However, if certain *m/z* values vary to a differing extent, the conventional *a* value approach might fail to capture the full variation or result in constraints that are too loose for certain *m/z*. For example, in the conventional *a* value approach, an *a* value of 0.3 was applied for all *m/z*'s while certain *m/z*'s could vary over 100%, failing to capture the variation of these *m/z*'s. But for certain *m/z*'s, an *a* value of 0.3 would result in too loose constraint given that the variation is less than 5%.

In this study, individual *m/z* was only allowed to vary within the range of the source profiles from different stoves (defined as the "limits" approach in SoFi (6.F1); https://datalystica.com/sofi, last access: 1 April 2021). In other words, different degrees of constraint were applied to individual *m/z* values to capture their specific variations instead of the universal constraint as in the conventional *a* value approach. To examine the statistical uncertainty of this approach, a bootstrap-based resampling strategy with a total of 100 runs was applied. Through bootstrapping, a set of new input matrix was created by random resampling of rows from the original ones (Paatero et al., 2014). By randomly duplicating some time points while excluding others, the original dimension of the input matrix was preserved. These ME-2 bootstrapping runs were averaged as the optimized solution, with the variation reflecting the model uncertainty.

## 3 Results and discussion

### 3.1 Source profiles for solid fuel burning in different domestic stoves

Table 1 summarizes the fuel types and stove types that were tested in the combustion experiments. Two scenarios are considered regarding the real-world application of solid fuel burning. One scenario is when people might purchase the same type of solid fuel (e.g., smokeless coal/biomass briquettes) from the same producers but have different stoves for heating their homes (i.e., the same batch of fuels burned in different stoves). The other scenario is when people might purchase the same type of fuel from different producers and burn them in different stoves (i.e., different batches of solid fuel burned in different stoves). Below, we discuss the signatures and differences of the profiles (i.e., organic aerosol mass spectra characterized with an ACSM), as well as their implications for OA source apportionment.

### 3.1.1 Biomass briquettes and smokeless coal burned in a conventional stove and an Ecodesign stove

Figure 1 compares the normalized organic aerosol mass spectra (MS) obtained from burning the same batch of biomass briquettes and smokeless coal in two different stoves – a conventional stove and an Ecodesign stove. Although sampled from

different stoves, the mass spectral signatures as a whole were similar with an uncentered $R^2$ (i.e., $\sim R^2$) of 0.87 for biomass briquettes and $\sim R^2$ of 0.97 for smokeless coal. All MS profiles appear to be dominated by fragments of $C_nH_{2n+1}$ (m/z 29, 43, 57, 71···) and $C_nH_{2n-1}$ (m/z 27, 41, 55, 69···), indicating a large contribution from saturated alkanes, alkenes, and/or cycloalkanes. However, the normalized peak intensities at specific m/z (e.g., m/z 41) sometimes varied significantly for the same type of solid fuel in the different stoves. The differences in the MS (discussed in Sect. 3.1.3) could be due to the different burning conditions (e.g., air supply and temperature) employed by the stoves, resulting in different thermal decomposition processes of the solid fuel and the corresponding pyrolysis products (Andreae, 2019; Weimer et al., 2008).

**3.1.2 Wood, peat, and smoky coal burned in a conventional stove and boiler stove**

Figure 2 compares the normalized organic aerosol mass spectra (MS) obtained from burning different batches of wood, peat, and smoky coal in a conventional stove (from this study) and a boiler stove (from Lin et al. (2017)). The wood-burning OA produced in the two stoves shows the largest variation with $\sim R^2=0.78$, followed by smoky coal ($\sim R^2=0.88$) and peat ($\sim R^2=0.95$). The large variation in the MS of wood burning was likely associated with the high volatile content (80.8% wt) in wood (Trubetskaya et al., 2021), which can be sensitive to the burning conditions. Although these solid fuels were purchased from different locations at different times (Dublin 2019 and Tipperary 2016) the general signatures were similar for each fuel type and displayed the expected marker ions. The key marker ion in wood burning OA appears at m/z 60 and m/z 73 (Alfarra et al., 2007). Mass fragment at m/z 60 (mostly from the $C_2H_4O_2^+$) is due to the fragmentation of anhydrosugars (e.g., levoglucosan, mannosan, and galactosan from the combustion of cellulose/hemicellulose; (Lee et al., 2010)) in the ACSM, and it is, therefore, commonly used as a marker for biomass burning in the AMS/ACSM studies (Cubison et al., 2011; Lee et al., 2010). Therefore, the differences in the content of cellulose/hemicellulose in the test fuels partly contribute to the differences in ion intensity at m/z 60 in ACSM. Specifically, the MS of wood burning OA has a prominent contribution from m/z 60 (i.e., *f60*>2.9%; *f60* denotes the fraction of m/z 60 in the total organic signal), while the MS of coal burning has a very low contribution from m/z 60 (*f60*<0.1%). In contrast, *f60* in the MS of peat burning was in between coal and wood (1.6-1.7%). This finding is consistent with the cellulose content in each of the solid fuels – wood>peat>>coal. While marker ions are important in the identification of specific OA factors during ambient studies, the differences in the intensities at specific m/z are an important source of uncertainty when used as inputs for ME-2 modelling (Canonaco et al., 2013; Canonaco et al., 2021). Therefore, examining variations in the intensities of specific m/z values due to the use of different stoves has great implications for factor analysis of an ambient dataset.

**3.1.3 Differences in source profiles and implications for factor analysis**

The MS obtained using different stoves are compared by plotting the relative differences of individual m/z values (calculated by $(f_{m/z,\text{ stove y}} - f_{m/z,\text{ stove x}}) / f_{m/z,\text{ stove x}}$ where $f_{m/z}$ represents the fraction of the measured m/z to the total organic signal, while stove y represents the Ecodesign or the Boiler stove, and stove x represents the conventional stove; Fig. S3 and S4). For wood burning in the conventional and boiler stoves, large differences (0.84 or 84%) were associated with the intensity of the marker

ion *m/z* 60 (Fig. S3). In addition to *m/z* 60, other fragments also showed large variations (Fig. S3). For example, the relative difference was 0.5 (or 50%) for *m/z* 44, a marker ion for aged or more oxidized OA (Canonaco et al., 2015). In ambient studies, the triangle space between *m/z* 44 and *m/z* 60 is often used to study the aging of biomass burning, in which a decreasing *f60* and an increasing *f*44 are usually associated with the atmospheric aging process (Ng et al., 2011a). However, the results from our study suggest that variations in *f60* and *f44* could also be due to different burning conditions (i.e., in different stoves) and

do not necessarily correspond to primary OA aging or atmospheric processing. For peat burning OA, there is a lower *f60* (0.016-0.017) than that for wood (0.029-0.053) due to the lower content of cellulose (Brown et al., 1988; Mikucioniene et al., 2019). Compared to wood, the peat MS appeared to be less affected by the stove type with a difference ratio of -0.06 (or 6%) for *m/z* 60 (Fig. 2). But for other fragments (e.g., *m/z* 29, 41,43…), a difference ratio of up to 0.46 is also indicative of significant variations caused by the type of stove.

*f60* in the MS of biomass briquettes was 0.005-0.008, which was 4-10 times lower than wood and 2-3 times lower than peat. This can be explained by the manufacturing process for the biomass briquettes, which involves torrefaction at a temperature of >250ºC that causes thermal decomposition of the raw biomass. The briquettes thus contain less cellulose and produce a lower *f60* as a result. The difference ratios for the MS of biomass briquettes burned in the conventional and Ecodesign stoves were in the range of -0.4 to 0.6 for the major fragments (e.g., *m/z* 41, 43, 55, 57). For some minor fragments (e.g., *m/z* 71 and

85), the difference ratios were even higher with values of up to 1.4 (i.e., 140%; Fig. S4), again suggesting the large impact of burning conditions on the MS profiles.

For the MS of smoky coal, *f60* was reduced to 0.00071-0.00081 (<0.1%) while for smokeless coal, *f60* was 0.0027-0.0045, both of which were lower than that for wood/peat. The reduced *f60* in the normalized mass spectra for smoky/smokeless coal is likely due to the breakdown of e.g., cellulose during coal formation over millions of years (Höök, 2012), resulting in a

215 relatively low content of cellulose, while accumulating other carbon-rich content, leading to the observed ions at other m/z's. As a comparison, the large contribution from the fragments at *m/z* 77, 91, and 115 suggests a high content of aromatic/polycyclic aromatic hydrocarbon (PAH) compounds in the smoky coal burning emissions. Specifically, *f77*, *f91*, and *f115* were in the range of 0.015-0.016, 0.014-0.015, and 0.019-0.026, respectively, for the MS of smoky coal. However, for the MS of smokeless coal, *f77, f91*, and *f115* were lower with values of less than 0.015. Compared to smoky coal, the lower

levels of aromatic/PAH-related fragments (i.e., m/z 77, 91, and 115) in the MS of smokeless coal are associated with its production process, which removed most of the volatiles in the raw coal during torrefaction at high temperatures (Trubetskaya et al., 2021), resulting in lower emission factors.

## 3.2 Use of different source profiles for source apportionment of organic aerosol in Dublin, Ireland

### 3.2.1 overview of ambient aerosol measurements

Figure 3 shows the time series of PM$_{2.5}$, PM$_1$ components, and OA factors (discussed in Sect. 3.2.2) in suburban Dublin from 1 November 2016 to 31 January 2017. During the sampling period, PM$_1$ (sum of ACSM and BC measurements) showed large

variations with 30 min averaged concentrations ranging from <0.5 µg m$^{-3}$ to 302.0 µg m$^{-3}$. In particular, PM$_1$ concentrations of 25 µg m$^{-3}$ were often (roughly 1 in 3 days) exceeded during the sampling period. The PM$_1$ at the sampling site showed a strong correlation with PM$_{2.5}$ at the Rathmines station (R$^2$ of 0.87, a slope of 0.74; Fig. S1), thus confirming that the ACSM sampling site was representative of the residential areas in southern Dublin (Fig. 3a). The slope of 0.74 for the linear relationship between PM$_1$ and PM$_{2.5}$ suggests PM$_1$ on average accounts for 74% of PM$_{2.5}$. However, during pollution events, the values of PM$_1$ and PM$_{2.5}$ are very similar, indicating most PM are in the submicron size ranges.

Consistent with the overall trend of PM$_1$, all the measured PM$_1$ components showed similar temporal variation with enhanced concentrations during the evening. The diurnal cycle of OA showed an increase from 16:00 (local time) which peaked during 20:00-22:00 (Fig. 4). However, OA decreased sharply overnight and remained at a low concentration during the day (8:00-16:00). Similarly, BC concentrations were over 4 times higher in the evening than during the day. The very similar diurnal patterns of OA and BC suggest they have common emission sources (i.e., heating) during the evening and night hours. Meteorological parameters like temperature and wind speed were also partly contributing to the elevated concentrations in the evening. The scatter plot (Fig. S5) between OA and temperature/wind speed suggests the high OA concentrations were coupled with low temperatures (<7 °C) and low wind speed (<5 m s$^{-1}$). Also, the shallower planetary boundary layer was an important factor for the increased OA concentration in the evening (Lin et al., 2018). In addition, the diurnal pattern of ammonium, nitrate, and sulfate all showed peaks at the same evening hours with the BC (Fig. 4a), suggesting they may also be related to heating emissions coupled with low temperatures in the evening (Fig. S6). Sulfate was likely associated with the primary emissions from solid fuel combustion given that, in the stove emission, sulfate was found to contribute to <1% of PM$_1$ for wood burning but the fraction of sulfate was up to 21% of PM$_1$ for smokeless coal burning, reflecting the higher content of sulfur in the raw fuel (Trubetskaya et al., 2021). This is consistent with our previous study (Lin et al., 2019), where we demonstrated that sulfate, nitrate, and ammonium can be locally emitted/formed, as well as regionally transported, through the comparison of the ACSM measurement at the same Dublin sampling site and at Carnsore Point, a regional background site.

### 3.2.2 Contribution of solid fuel burning to ambient organic aerosol

To evaluate the contributions of solid fuel burning to the ambient OA in suburban Dublin, the MS of wood, peat, and smoky coal were used as the anchoring profiles for ME-2 modelling (Lin et al., 2017; Lin et al., 2018; Trubetskaya et al., 2021). In the ME-2 analysis, the individual *m/z* values in the MS for wood, peat, and coal were allowed to vary between the reference profiles (i.e., the limits approach (see Sect. Method); Fig. S7-S10). The time series of solid fuel burning factors (Figure 3c) were very similar (i.e., all peaked during the same evening hours; Fig. S8) due to the similar emission time from the domestic heating activities. This is the reason why ME-2 was used to separate these factors since unconstrained PMF led to highly mixed and non-physically meaningful factors (Lin et al., 2017). In addition to solid fuels, a hydrocarbon-like OA (HOA) factor and an oxygenated OA (OOA) factor were also resolved (Fig. S7). HOA was associated with the emissions from oil heating during the evening while OOA was related to regional transport and/or secondary processes (Lin et al., 2020). Increasing the number of factors during the ME-2 analysis (i.e., the 6-factor solution; Fig. S9) identified an additional OOA factor (OOA2) which,

however, featured a very low signal at $m/z$ 41 in the normalized mass spectra but a similar signal level at $m/z$ 43 with the already identified OOA factor (Fig. S7). The unambiguous separation of two OOA types requires further research. Nevertheless, for the 5-factor solution, the good correlation ($R^2=1$, slope = 0.99; Fig. S10) between the time series of the explained fraction and the PMF input suggests the 5-factor solution explained the input matrix well.

Figure 4b shows the diurnal cycle of the averaged contribution of the resolved factors over the entire period. On average, solid fuel burning (the sum of peat, wood, and coal) was the major contributor (>50%) to the total OA during the evening, while during the day, OOA was the dominant factor. Therefore, primary emissions from solid fuel burning were the dominant sources of pollution in the evening, while regional transport and/or secondary processes of OA were the major source during the day. The oil heating factor was contributing, on average, 22-25% of the total OA in the evening. Even though the overall results from the limits approach were consistent with those from the conventional $a$ value approach in Lin et al. (2018), a detailed comparison of results between the two approaches as well as the corresponding uncertainties are provided below.

### 3.2.3 Comparison of OA source apportionment using the limits approach versus the $a$ value approach

Figure 5 shows the comparison of the time series of OA factors resolved by the limits approach and the conventional $a$ value approach (Lin et al., 2018), while Table 2 shows the corresponding uncertainties. The statistical uncertainty of the limits approach was evaluated through the bootstrap-based resampling strategy (See Method section), while the model uncertainties for the $a$ value approach was the variation (one standard deviation) of the accepted ME-2 solutions with the combination of different $a$ values (Lin et al., 2018). The model uncertainty for the limits approach was in the range of 2-7%, considerably lower than 8-29% for the $a$ value approach. The low uncertainty for the limits approach suggested the solution was relatively stable. In contrast, the relatively large uncertainty for the $a$ value approach suggested the degree of variation from the anchoring profiles could cause uncertainties in the solution of up to 29%. In addition to the model uncertainties, the dilution and cooling of the aerosol samples after mixing in the ambient atmosphere, as well as atmospheric processes (e.g., night-time chemistry with $NO_3$ radical (Kiendler-Scharr et al., 2016)) are also important sources of uncertainties in OA source apportionment since these factors could also cause variation in the mass spectra. In this study, our combustion experiment deployed a Dekati Diluter (See Method section) to simulate the dilution and cooling of the raw flue gas samples through mixing with compressed clean air.

Overall, the time series of the OA factors for peat, HOA, and OOA were well correlated with $R^2 > 0.95$ and slopes in the range of 0.95-1.10 (Figure 5), suggesting excellent agreement between the two approaches despite the difference in uncertainties (Table 1). Although the wood burning factor time series from the two approaches had a high correlation coefficient ($R^2$ of 0.99), a slope of 1.33 indicated the quantification of the OA factor of wood burning varied to a larger extent (i.e., 33%). In contrast, the OA factor for coal burning showed the poorest correlation with an $R^2$ of 0.37 and a slope of 0.44. The poor correlation for the coal burning factor was likely due to the low contribution to the total OA (<10%; Fig. 4c), and the large uncertainty from the $a$ value approach (29%; Table 1). Nevertheless, both approaches pointed to an important source of solid fuel burning in winter Dublin, with the sum of peat, wood and coal factors, on average, contributing over 50% of the

total OA during the evening hours (20:00-23:00). Specifically, both approaches showed peat burning being the largest OA factor (39% (Fig. 5f) vs 41% (Fig. 5g)), followed by HOA (24 vs 25%), OOA (20 vs 18%), wood (14% vs 11%), and coal (4% vs 5%) during the evening hours (20:00-23:00) when stove emissions dominate. Therefore, high variations in specific $m/z$ contributions to OA mass spectra from different fuel and stove types do not translate into high source apportionment uncertainties owing to the robust ME-2 approach. Moreover, the time series of OOA showed spikes concurrent with primary factors (Fig. S8) during the evening and night-time, suggesting OOA was probably associated with the condensation of semi-volatile species and/or aging of primary emissions in the real atmosphere (Tiitta et al., 2016). As a result, the contribution from solid fuel burning could be higher than solely represented by the POA fraction given that OOA, on average, accounted for approximately 20% of the OA in the evening (Fig. 4b).

### 3.3 Atmospheric implications

Our results indicate the emission profiles, in terms of the specific $m/z$ values in the organic mass spectra, varied significantly even for the same type of solid fuel burned in different types of stoves. Taken into account such variations, the uncertainties in the source apportionment of ambient organic aerosol were reduced. This study provides better constraints in the contribution of solid fuel burning to the ambient organic aerosol and is of importance for public health and policymakers considering that it is often the case that different batches of solid fuels are often burned in different stoves in real-world applications. In particular, solid fuels of peat and wood, both of biomass nature, were found to contribute to a considerable fraction (>50%) of the total organic matter. However, according to the Central Statistical Office in Ireland (CSO, 2016), only a small number (<10%) of households use peat and wood as the primary heating source, with the majority (>90%) using the relatively clean energy of gas and electricity. Trubetskaya et al. (2021) showed peat (38-92 g GJ$^{-1}$) and wood (44-179 g GJ$^{-1}$) had higher emission factors than smoky coal (17-29 g GJ$^{-1}$), smokeless coal (5-18 g GJ$^{-1}$), biomass briquettes (7-28 g GJ$^{-1}$). Therefore, despite the small use of peat and wood, their high emission factors make these fuels important factors driving the pollution events observed during the heating season. Moreover, the good correlation between the time series of PM$_1$ and PM$_{2.5}$, despite the distance of 3 km between the two measurements, suggests the pollution events covered a large area in Dublin with a spatial scale of at least 3 km in radius. In other words, the air quality for those using clean energy of gas and electricity was also impaired by the small group of people using peat and wood.

Biomass burning is a carbon neutral energy source given that biomass captures almost the same amount of carbon dioxide (CO$_2$) through photosynthesis during growing as is released when biomass is burned (Marland, 2010). This makes biomass an alternative to fossil fuels to combat climate change. Replacing fossil fuels with biomass may result in lower CO$_2$ emissions overall. However, in terms of particulate emission, burning biomass can cause serious air pollution as shown in this study. In other European sites, biomass burning has also been reported to be an important source of particulate pollution (Alfarra et al., 2007; Allan et al., 2010; Crippa et al., 2014). Therefore, rather than promoting the use of biomass burning, new emission controls on the residential biomass burning e.g., through the introduction of more energy efficient and low emission stove

(Trubetskaya et al., 2021), is needed to improve the overall air quality. In 1990, the Irish government introduced a ban on the marketing, sale, and distribution of bituminous (smoky) coal in Dublin. This led to a 70% reduction in the average black smoke levels during the post-ban period compared to the pre-ban period (Goodman et al., 2009). Consistently, our results showed coal combustion accounted for a small fraction (<5%) of the organic mass. These results suggest appropriate intervention can
be effective at reducing particulate pollution. Therefore, extending the ban to the use of peat and wood is expected to further improve the air quality in Ireland.

## 4 Conclusion

This study has provided a detailed characterization and comparison of organic aerosol mass spectra produced from burning a range of solid fuels in several stoves. Key ions (e.g., *m/z* 60) remain important markers for identifying solid fuel emissions
using ACSM data. However, the intensities at different *m/z* values, including the marker ions, varied significantly from <5% to >100%, and are an important source of uncertainties when using their respective mass spectra as anchoring profiles in the conventional *a* value approach in ME-2 modeling. Using the limits approach in ME-2 analysis, the contributions of peat, wood, and coal to the ambient OA were evaluated and compared with this conventional *a* value approach. The ME-2 solution was stable for the limits approach with uncertainties in the range of 2-7%, while relatively large uncertainties (8-29 %) were found
for the *a* value approach. The peat burning factor was subject to fewer uncertainties and showed a good agreement between the two approaches ($R^2$ of 0.99 and a slope of 0.96), while wood and coal OA factors showed a relatively larger variation with a slope of 1.33 and 0.44, respectively. Both approaches showed that coal burning was contributing <10% of the ambient OA, while peat and wood contributed substantially (>50%) to the ambient OA in the evening hours of the heating season despite their small uses. The results from this study suggest locally obtained reference source profiles, in combination with robust ME-
2 approach, can reduce the uncertainty and, therefore, are better for quantitative source apportionment of primary emissions from solid fuel burning. The finding from this study holds important implications for public health and policymakers considering that it is often the case that different batches of solid fuels are often burned in different stoves in real-world applications.

## Data Availability

All data needed to evaluate the conclusions in the paper are present in the paper and/or the Supplementary Materials. Also, all data used in the study are available from the corresponding authors upon request.

**Author Contribution**

JO and CL conceived and designed the experiments; CL, JO, DC, WS and AT performed the measurements; CL, DC, WX, and JO analyzed the data; CL prepared the manuscript with input from all co-authors.

**Competing interests**

The authors declare that they have no conflict of interest.

**Acknowledgments**

This work was supported by EPA Ireland (AEROSOURCE, 2016-CCRP-MS-31) and Department of Environment, Climate and Communications. CL acknowledges the support from the Strategic Priority Research Program of Chinese Academy of Sciences (Grant No. XDB40030202), the National Natural Science Foundation of China (NSFC) under Grant No. 4210072504, and the Institute of Earth Environment Chinese Academy of Sciences (no. E051QB2837). The authors would also like to acknowledge the contribution of the COST Action CA16109 (COLOSSAL) and MaREI, the SFI Research Centre for Energy, Climate and Marine.

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

**Table 1. Fuel types and stove types that were included in the combustion experiments.**

| Fuel type | Stove type | | |
|---|---|---|---|
| | Conventional | Ecodesign | Boiler |
| Smokeless coal | √ | √ | |
| Biomass briquettes | √ | √ | |
| Peat | √ | | √* |
| Wood | √ | | √* |
| Smoky Coal | √ | | √* |

*A different batch of fuel was tested.

**Table 2. Uncertainties in OA factor attribution obtained using the limits approach versus the *a* value approach.**

| | ME-2 model uncertainties | |
|---|---|---|
| OA factors | Limits approach | *a* value approach |
| Peat | 2% | 11% |
| Wood | 2% | 15% |
| Coal | 7% | 29% |
| HOA | 2% | 12% |
| OOA | 2% | 8% |

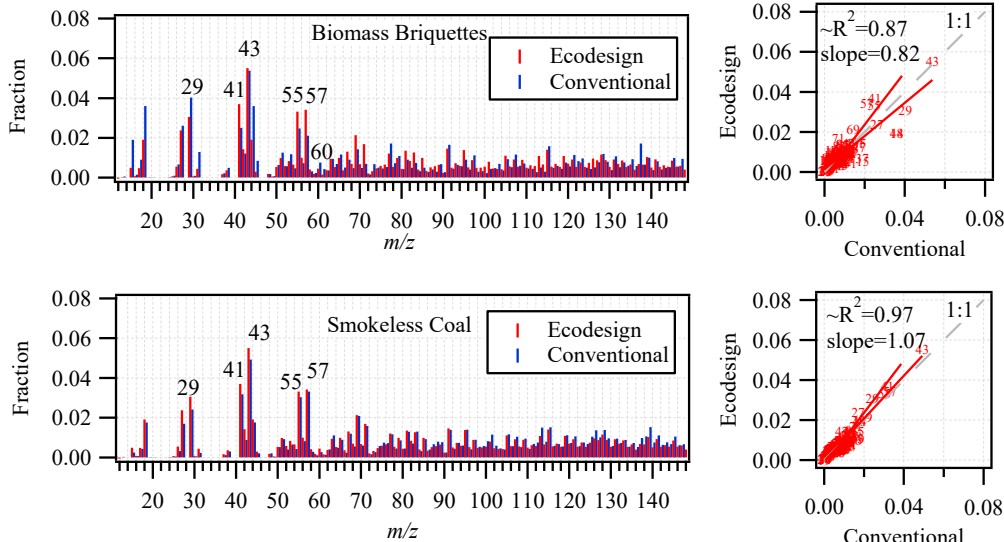

**Figure 1.** Source profile (i.e, mass spectra; left panel) of the organic aerosol from the combustion of biomass briquettes, and smokeless coal in the conventional versus Ecodesign stove, and their corresponding linear correlation relationship (right panel). For Clarity, *m/z* values in the mass spectra from the Conventional stove were offset by 0.5. Inset text shows the uncentered $R^2$ (i.e., $\sim R^2$) and the slope of the correlation.

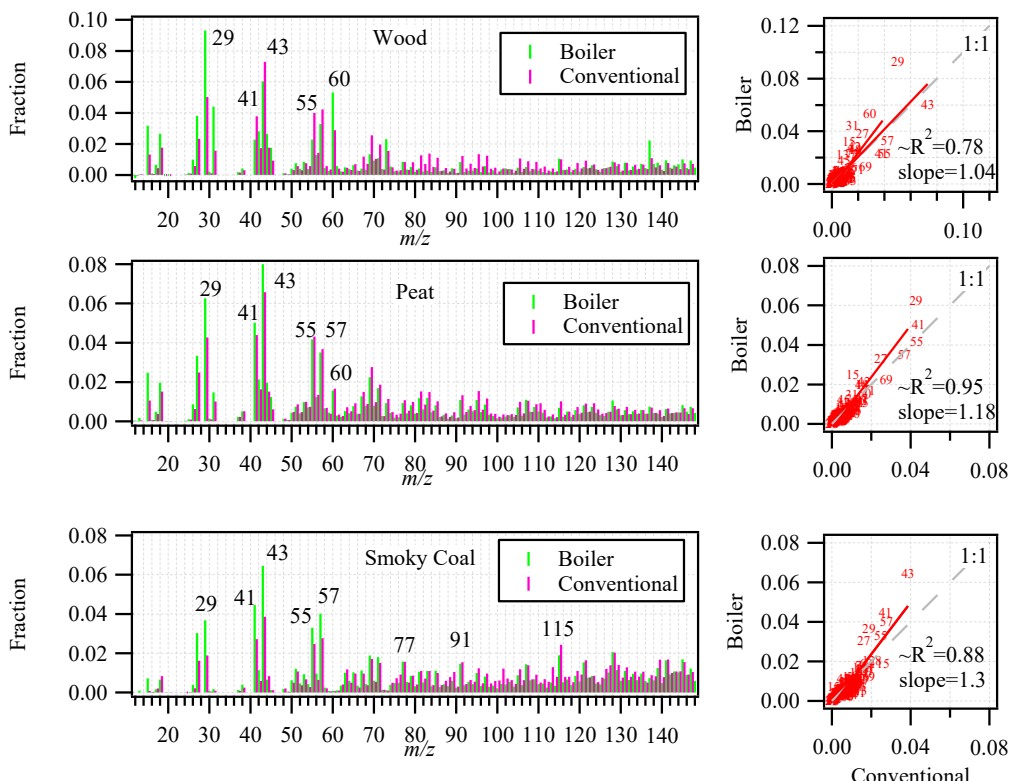

**Figure 2.** Source profile (i.e, mass spectra; left panel) of the organic aerosol from the combustion of wood, peat, and smoky coal in the boiler versus the conventional stove, and their corresponding linear correlation relationship (right panel). For Clarity, *m/z* values in the mass spectra from the Conventional stove were offset by 0.5. Inset text shows the uncentered $R^2$ (i.e., ~$R^2$) and the slope of the correlation.

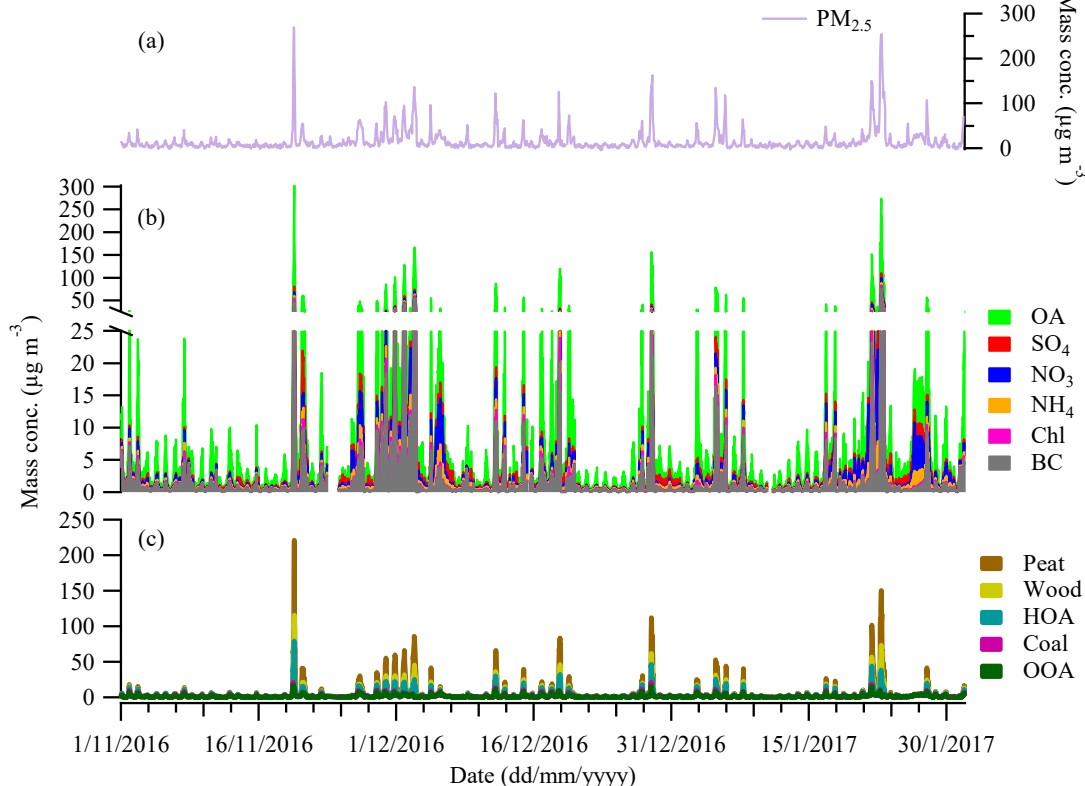

**Figure 3.** Time series of (a) PM$_{2.5}$ at Rathmines station, ~3 km west of the ACSM sampling site; (b) organic aerosol (OA), sulfate (SO$_4$), ammonium (NH$_4$), nitrate (NO$_3$), chloride (Chl) and black carbon (BC); and (c) OA factors for peat, wood, HOA, coal, and OOA obtained using the limits approach.

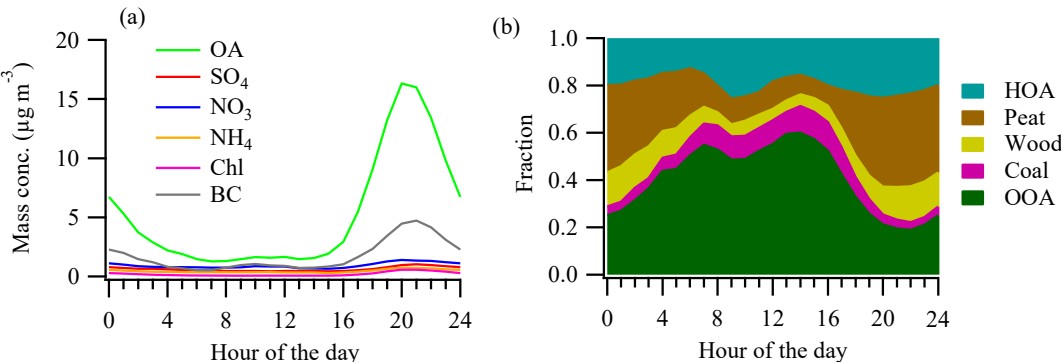

**Figure 4.** Averaged diurnal cycle of (a) organic aerosol (OA), sulfate (SO$_4$), ammonium (NH$_4$), nitrate (NO$_3$), and chloride (Chl) and black carbon (BC); (b) relative contribution of OA factors of HOA, peat, wood, coal, and OOA.

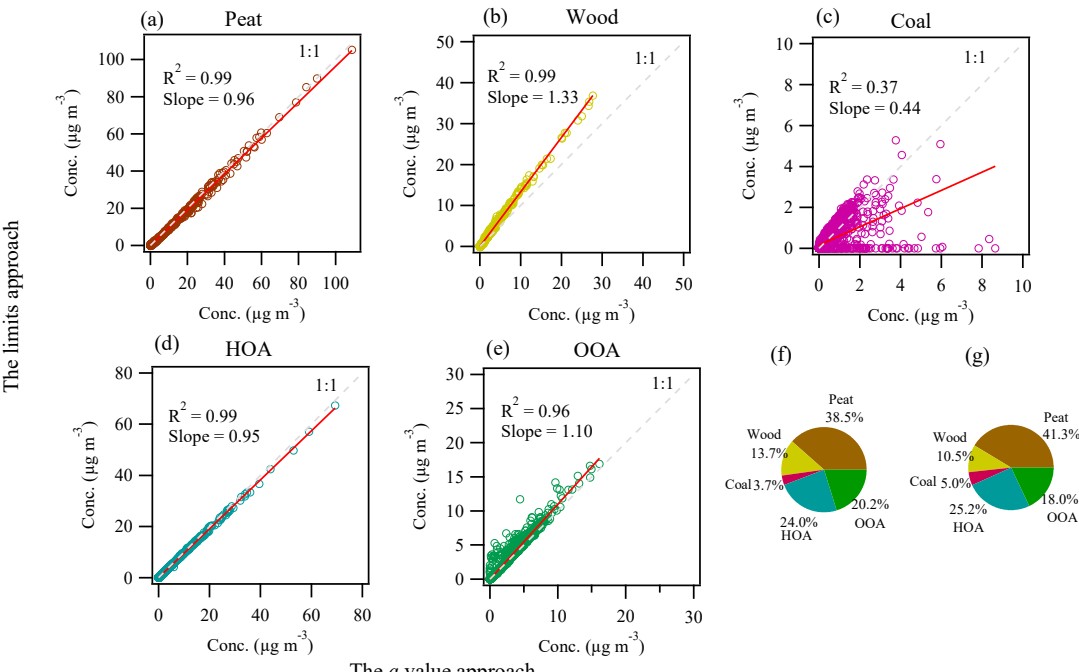

**Figure 5. Scatter plot of the time series of OA factors of (a) peat, (b) wood, (c) coal, (d) HOA and (e) OOA resolved by the limits approach versus the conventional *a* value approach; Averaged relative contribution of the resolved factors during the evening hours (20:00-23:00) by (f) the limits approach and (g) the *a* value approach. The dash grey line in the scatter plot is the 1:1 reference line, while the red line is the linear regression with $R^2$ and slope values shown on top.**