# Peer review of "On the use of reference mass spectra for reducing uncertainty in source apportionment of solid fuel burning in ambient organic aerosol"

_Atmospheric Measurement Techniques, 2021_

## Author Comment (AC1)

We thank the reviewers for their careful examination of our manuscript, and the insightful comments which have helped to improve our manuscript substantially. Below we provided a point-to-point response to the reviewers' comment, where the reviewers' comments are in black, and our responses are in blue.

Reviewer #1

This study presents laboratory studies testing various solid fuels and stoves as well as ambient measurements investigating solid fuel emissions using the limits approach within ME2. The findings in this research will help improving the source apportionment of OA by applying the limits approach within ME2 analysis.

The paper, which fits well within the scope of AMT, is recommended to be published after working on the following main comments.

Response: We thank the reviewer for the positive comment. See below for a point-to-point response to the comment.

The introduction is well elaborated with adequate use of references. However, it may be improved by describing more Dublin, previous solid fuel studies and the importance of the present work.

Response: We have now provided more discussions on our previous Dublin studies in the Introduction Section and highlighted the importance of the present work.

It now reads, "Dublin is a moderately sized city in western Europe with a population of around 1 million. Recent studies in Dublin show that residential burning of solid fuels – mainly peat and wood, but also coal to a lesser degree – is a significant source of ambient organic aerosol (OA) during the heating season (Lin et al., 2019; Lin et al., 2018). In a case study, Lin et al. (2018) show residential heating and particularly peat and wood burning caused an extraordinarily high concentration (over 300 $\mu g\ m^{-3}$) of submicron aerosol, affecting air quality on a local to regional scale in suburban Dublin. Source attribution of the measured OA to different types of solid fuels was performed using reference profiles from locally sourced fuels (Lin et al., 2017) as the anchoring profiles in the ME-2 modelling (i.e., the *a* value approach (Canonaco et al., 2013)). The reference profiles for solid fuels were obtained from a combustion experiment using a boiler stove with no emission control (Lin et al., 2017). However, the question remains on how these reference profiles vary with stove type and what uncertainties this variation causes in the ME-2 modelling."

I think This work needs details about the ME2 analysis and the selection of the optimal solution, perhaps explain it in the supplement material. The authors mention ME2 does a better job compared to PMF and also that the limits approach improves the ME-2 source apportionment. However, there are no details on the factor selection. Also, the R2 is not the best statistical parameter to use when comparing mass spectra due to the large contribution of the important m/z for example m/z 55 or m/z 57. A more suitable parameter should be used, for example uncentered R2 when analysing mass spectra.

Response: We have now provided more details about ME2 analysis and the selection of the optimal solution in the supplementary. Specifically, to show that ME2 does a better job compared to PMF, we performed unconstrained or free PMF analysis. As shown in Figure R1 (or Figure S2 in the revised supplementary), the diurnal pattern of the free PMF factors all showed elevated concentration in the evening and night, suggesting heating-related sources coupled with a shallow boundary layer. However, for the free PMF solution, mixing between factors is evident given that the mass spectra for some of the factors were suffering from the missing of some important m/z's such as m/z 43, 44 (Figure R1), which

was attributed to other factors (i.e., mixing with other factors). To reducing the mixing, ME-2 was applied to constrain the reference mass spectra of the potentially contributing sources (i.e., the heating oil, peat, wood, and coal). Oil, peat, wood, and coal were constrained because these fuels are important residential heating sources according to the Sustainable Energy Authority of Ireland (SEAI, 2018) and the Central Statistics Office (CSO, 2016) in Ireland. Consistently, previous air quality studies in Ireland have shown their important role in contributing to the organic aerosol during the winter period (Dall'Osto et al., 2013; Kourtchev et al., 2011; Lin et al., 2020).

For the ME-2 (i.e., the constrained PMF) solution, the 5-factor solution was deemed as the most optimized solution, including the 4 constrained factors (i.e., HOA, peat, wood, and coal), and one unconstrained factor (i.e., OOA). Note that OOA was also resolved by free PMF and only one type of OOA was resolved because increasing the number of ME-2/PMF factors does not lead to new interpretable OOA factors. The 5-factor solution was also consistent with our previous study using the conventional $a$ value approach (Lin et al., 2018).

In terms of correlation parameters for the mass spectra, we agree that $R^2$ could be influenced by large signals like m/z 55 and m/z 57. We have now used the parameter of the uncentered $R^2$ (i.e., $\sim R^2$) when comparing mass spectra in the revised manuscript. The overall trend of $\sim R^2$ remains in the same order, and, therefore, the overall conclusion is the same with $R^2$. Related figures and text have been replaced by $\sim R^2$. (See Figure R2 and R3 or Figure 1 and Figure 2 in the revised manuscript)

[Figure]

Figure R1. Mass spectral profiles and diurnal, as well as the relative contribution of the free PMF four- (top) and five- (bottom) factor solutions over the entire period. The mass spectra for some of the factors were suffering from the missing of some important m/z's such as m/z 43, 44, which was attributed to other factors (i.e., mixing with other factors).

[Figure]

Figure R2. Source profile (i.e, mass spectra; left panel) of the organic aerosol from the combustion of biomass briquettes, and smokeless coal in the conventional versus Ecodesign stove, and their corresponding linear correlation relationship (right panel). For Clarity, $m/z$ values in the mass spectra from the Conventional stove were offset by 0.5. Inset text shows the uncentered $R^2$ (i.e., ~$R^2$) and the slope of the correlation.

[Figure]

Figure R3. Source profile (i.e, mass spectra; left panel) of the organic aerosol from the combustion of wood, peat, and smoky coal in the boiler versus the conventional stove, and their corresponding linear correlation relationship (right panel). For Clarity, $m/z$ values in the mass spectra from the Conventional stove were offset by 0.5. Inset text shows the uncentered $R^2$ (i.e., ~$R^2$) and the slope of the correlation.

The authors mention the technical details of the lab experiments are in a previous publication but perhaps a quick overview should be covered here, i.e. the sampling times/conditions, maybe diagrams, photos, etc.

Response: We have now provided a quick overview of the combustion experiments as well as diagrams and photos (See Schematic R1 and Schematic S1 in the supplementary).

It now reads, "…Five fuel types were tested including wood, peat, smoky coal, biomass briquettes, and smokeless coal (Table 1). Specifically, wood logs were cut from softwood grown in Ireland; peat was obtained from the peatland in Leitrim, Ireland, and was naturally dried before testing; smoky coal (Silesia, Poland) were purchased from local retail outlets (Trubetskaya et al., 2021); biomass briquettes and smokeless coal (i.e., Ecobrite ovoids) were manufactured at Arigna Fuels (Carrick on Shannon, Ireland)…

…The combustion experiment lasted 1-3 hours depending on the fuel and stove types. The time resolution of ACSM was set to 2 min to capture the variation of the combustion emission…"

[Figure]

Schematic R1. Schematic of the combustion experiment set-up. The conventional stove and the Ecodesign stove (Trubetskaya et al., 2021) were alternatively tested.

Line 17. Perhaps change ACSM by aerosol mass spectrometers. Mass spectra are used for source apportionment with all the AMS instruments not only ACSM.
Response: Changed.

Line 19. When the authors mention stoves, I think this refers only to solid fuel burning used for cooking, is that correct? If that is the case how to differentiate between solid fuel and biomass burning? Or solid fuel used for other activities apart from cooking?
Response: Here we refer to the heating stove (See Schematic R1). Solid fuels are mostly used for heating in Ireland (NAEI, 2017). Also, the diurnal pattern of the heating emissions shows a typical increase in the evening/nighttime, which was observed frequently in this study, while we cannot find a lunchtime meal peak from cooking emissions. Therefore, cooking emissions are a minor contributor to the measured PM in this study. We have specified this in the revised text.

Line 23 how can the authors attribute a >100% variation in the m/z? where in the manuscript is that mentioned?

Response: It is now mentioned in Sect. 3.13. It reads, "…some minor fragments (e.g., *m/z* 71 and 85), the difference ratios were even higher with values of up to 1.4 (i.e., 140%; Fig. S2), again suggesting the large impact of burning conditions on the MS profiles."

Line 29 What do the authors refer as "despite their small uses"?   are you talking about the ME2 approaches or the solid fuels?

Response: We were talking about the use of peat and wood was small when compared to the use of electricity and gas in Dublin. It now reads, "… despite their small uses compared to electricity and gas…"

Line 44 What is the purpose of the reference "Chen et al., 2020"? I think the references of Canonaco and Paatero you cover the ME2 and PMF nicely.

Response: The reference of Chen et al., (2020) is now removed.

Line45 I would not be as strong in the statement to say that "ME2 is a significant source of uncertainty" at least not using references 9 years old or more. You might want either to rephrase the line mentioning something like "ME2 can be a significant …" or find more recent references.

Response: We agree that this would be a strong statement. We have toned down the statement as suggested. Now changed to "…can be a significant source of uncertainty"

Line 70 It would be good to see a diagram and/or photos of the stoves.

Response: Please see the reply to the previous comment above. We have now added a schematic and photos of the stove in the revised supplementary.

Line 82 Was the ACSM calibrated? If so, please describe it.

Response: We have now described the ACSM calibration. It now reads, "ACSM was calibrated following the procedure described by Ng et al. (2011). Briefly, a Scanning Mobility Particle Sizer (SMPS, TSI 3938) was used to size-select (300 nm) the atomized ammonium nitrate or ammonium sulfate, which was subsequently fed into ACSM system."

Line 92 It might be a good idea to add a map, perhaps in the supplement, indicating the location of the sampling site and showing the area of Dublin for readers that are not familiar with the location. Perhaps to extend a bit more in the site description.

Response: We have now added a map (adapted from Google Maps) in the supplement (See Figure R4 below or Figure S1 in the supplement) and have provided more description on the sampling site. It now reads, "The ACSM sampling site is ~5 km south to the Dublin city center and is ~500 m away from the nearby road (Fig. S1). ACSM measurements were conducted on the roof of the O'Brien Centre for Science building ($\sim$ 30 m above the ground). Previous studies conducted at the same sampling site show the aerosol population was mainly affected by the heating emissions but with a relatively minor contribution from traffic or cooking emissions (Lin et al., 2020; Lin et al., 2018)."

[Figure]

Figure R4. (a) Sampling site for PM$_1$ at UCD and the PM$_{2.5}$ measurement site at Rathmines marked by the red cycles; (b) scatter plot between UCD PM$_1$ and Rathmines PM$_{2.5}$. Also shown in (b) are the correlation (R$^2$), slope, and intercept for the linear relationship. The map is adapted from Google Maps.

Line 97 Please add the website link and date of last access.

Response: Website link and date of last access (www.airquaity.ie; Last access: 1 September 2021) is now added in the revised text.

Line 104 I kind of follow the description of the PMF model equation because I am familiar with the topic. However, for someone else might be challenging to understand it as it is. Please rephrase it, for instance what does "p" represent? Or use the other equation showing the summatory and the use of I, j, p, etc.

Response: Sorry for the confusion. We have now rephrased this equation, it now reads, "The PMF model in matrix notation is defined as:

$$X = GF+E,$$

where the measured matrix X is approximated by the product of G and F, while E is the model residual."

Line 112 It seems that there have been a couple of solid fuel studies in Dublin. Those can be used in the introduction to inform the reader about the current findings and identifying the needs of the current study.

Response: Please also see the reply to the comment above. Relevant studies are cited and discussed in the revised manuscript. It now reads, "Dublin is a moderately sized city in western Europe with a population of around 1 million. Recent studies in Dublin show that residential burning of solid fuels – mainly peat and wood, but also coal to a lesser degree – is a significant source of ambient organic aerosol (OA) during the heating season (Lin et al., 2019; Lin et al., 2018). In a case study, Lin et al. (2018) show residential heating and particularly peat and wood burning caused an extraordinarily high concentration (over 300 µg m$^{-3}$) of submicron aerosol, affecting air quality on a local to regional scale in suburban Dublin. Source attribution of the measured OA to different types of solid fuels was performed using reference profiles from locally sourced fuels (Lin et al., 2017) as the anchoring profiles in the ME-2 modelling (i.e., the *a* value approach (Canonaco et al., 2013)). The reference profiles for solid fuels were obtained from a combustion experiment using a boiler stove with no emission control (Lin et al., 2017). However, the question remains on how these reference profiles vary with stove type and what uncertainties this variation causes in the ME-2 modelling."

Line 114 I think Paatero 1999 (https://doi.org/10.1080/10618600.1999.10474853) or Canonaco et al

2013 would be more suitable references here.

Response: Canonaco et al., (2013) is now cited instead of Lin et al., (2017)

Line 115 Again, I think here the authors make a strong statement that ME2 is better than PMF by involving mass spec from previous studies and it is assuming the PMF solution will be with mixed inaccurate factors. I think the authors should rephrase this paragraph mentioning the caveats of using inaccurate target profiles and also that PMF does a good job on performing source apportionment and is only when the target profiles are mixed when the use of ME2 shows an improved performance if the user does an extensive and objective analysis.

Response: We have now rephrased this statement to mention the suggested caveats of using PMF and ME-2. It now reads, "As shown in Fig. S2, unconstrained or free PMF suffered from factor mixing due to temporal covariation of the candidate factors (i.e., all increasing in the evening corresponding to the time of domestic heating activities). To evaluate the contribution of different types of solid fuels, source profiles obtained from the combustion experiments can be used as the anchoring factor-profiles (i.e., reference mass spectra) in the ME-2 algorithm (Canonaco et al., 2013). However, without extensive and objective analysis, both free PMF and ME-2 analysis can fail to apportion the sources accurately especially when the reference mass spectra can be complicated by the use of different fuels, stoves, and burning conditions."

Line 121 This is why the user should do an intensive analysis with ME2 testing different a-values and/or different mass spec target profiles.

Response: Agree. This is part of the motivation for this study.

Line 125 is this "SoFi" the same SoFi version 6.F1 mentioned in line 101? Or did the authors used different versions?

Response: Same versions. It now reads, "…the "limits" approach in SoFi (6.F1)"

127 What is a bootstrap-based resampling strategy? Could you explain it in a couple of lines?

Response: We have now explained the bootstrap strategy. It now reads, "Through bootstrapping, a set of new input matrix was created by random resampling of rows from the original ones (Paatero et al., 2014). By randomly duplicating some time points while excluding others, the original dimension of the input matrix was preserved."

134 What I understand here is that the stoves are used for heating and not for cooking. Please, explain this in the introduction so the reader could follow the manuscript nicely.

Response: Now explained. It now reads, "…in two different heating stoves…"

Line 165 Apart from m/z 60, m/z 73 is also a solid fuel marker. The authors might want to consider include it into the analysis.

Response: We agree that m/z 73 is also elevated in the biomass burning emissions, and therefore can be used as a marker in addition to m/z 60 (Alfarra et al., 2007). However, since m/z 60 is mostly from the $C_2H_4O_2^+$ ion due to the fragmentation of anhydrosugars in ACSM/AMS, and Cubison et al. (2011) proposed to use the level of *f60* (the fraction of m/z 60 in the total organic signal) to indicate the presence/absence of biomass burning, we tend to focus on discussing m/z 60 in the revised text.

It now reads, "The key marker ion in wood burning OA appears at *m/z* 60 and m/z 73 (Alfarra et al., 2007). Mass fragment at m/z 60 (mostly from the $C_2H_4O_2^+$) is due to the fragmentation of anhydrosugars (e.g., levoglucosan, mannosan, and galactosan from the combustion of cellulose/hemicellulose; (Lee et al., 2010)) in the ACSM, is, therefore, commonly used as a marker for biomass burning in the AMS/ACSM studies (Cubison et al., 2011; Lee et al., 2010)."

Line 166 Are fm/z calculated the same as f60 and f44 in line 170? If not, maybe use different acronyms.
Response: We have now clarified $f_{m/z}$ are calculated the same as *f60* and *f40*. It now reads, "…(calculated by $f_{m/z, stove\,y} - f_{m/z, stove\,x}$) $/f_{m/z, stove\,x}$ where $f_{m/z}$ represents the fraction of the measured m/z to the total organic signal, while stove y represents the Ecodesign or the Boiler stove, and stove x represents the conventional stove…"

Line 169 I think Ng et al (2011) (doi:10.5194/acp-11-6465-2011) would be a more appropriate reference here to avoid confusion.
Response: The original citations are now replaced by Ng et al., (2011) in the revised text.

Figure S1 in supplement. The Relative difference refers to fm/z difference or m/z difference? The authors might want to explain figure S1 in supplement, for instance who is y and who is x based on equation in line 166.
Response: The relative difference refers to $f_{m/z}$ difference. Also, y and x are now explained (see the reply to the previous comment or the caption in Figure S1 (now Figure S3) and Figure S2 (now Figure S4))

Line 189 What are these "PAH-related fragments", do you mention them previously?
Response: PAH-related fragments is now changed to "…aromatic/PAH-related fragments (i.e., *m/z* 77, 91, and 115)…"

Line 203 How could the authors state that solid fuel burning emissions are the dominant source due to OA and BC similar diurnal trend? At this stage is why we use PMF/ME2, to identify potential sources and do not to conclude from OA diurnal cycles.
Response: Agree. Now changed to "The very similar diurnal patterns of OA and BC suggest they have common emission sources (i.e., heating) during the evening and night hours"

Line 204 SOA can be transported from long distances and/or to increase concentrations due to drop in the boundary layer height and not necessarily to represent the contribution of heating emissions.
Response: We agree that the collapse of the boundary layer can increase the concentrations of SOA. However, the diurnal pattern of OOA was similar to the POA factors (see Figure R5 or Figure S8 in the supplementary), suggesting most OOA in the evening was from heating emissions (condensation of semi-volatile organic compounds or through night-time chemistry (Tiitta et al., 2016))

[Figure]

Figure R5. Averaged diurnal cycle of the ME-2 5-factor solution. Error bar represents one standard deviation.

Line 195 300 μg m-3 are high aerosol concentrations it would be interesting to know the type of monitoring site, the type of city Dublin is, I guess Nov-Jan is considered to be Winter and this makes sense for analysing solid fuel OA concentrations but it would be useful to explain all this maybe in the introduction. Also, it would be interesting to analyse the large peaks and see if they can be explain i.e by looking at the meteorology or special events. Did the authors look at the boundary layer height for this manuscript?

Response: We have now provided more details about the sampling site in Dublin both in the revised Introduction and Method section (also see the reply above). The case study on the pollution event with 300 μg m$^{-3}$ is already detailed in our previous study (Lin et al., 2018). Also, the meteorological parameters and boundary layer height impacts are discussed in Lin et al. (2018). In this study, we focus on the impacts of the variation of the reference profiles on the ME-2 source apportionment, and, therefore, tend not to repeat what was already studied in our previous studies. In the Introduction section, it now reads, "…Dublin is a moderately sized city in western Europe with a population of around 1 million. Recent studies in Dublin, Ireland show that residential burning of solid fuels – mainly peat and wood, but also coal to a lesser degree – is a significant source of ambient organic aerosol (OA) during the heating season (Lin et al., 2019; Lin et al., 2018). In a case study, Lin et al. (2018) show residential heating and particularly peat and wood burning caused an extraordinarily high concentration (over 300 μg m$^{-3}$) of submicron aerosol, affecting air quality on a local to regional scale in suburban Dublin. Source attribution of the measured OA to different types of solid fuels was performed using reference profiles from locally sourced fuels (Lin et al., 2017) as the anchoring profiles in the ME-2 modeling (i.e., the a value approach (Canonaco et al., 2013)) The reference profiles for solid fuels were obtained from a combustion experiment using a boiler stove with no emission control (Lin et al., 2017)., peat and wood burning were found to contribute over 50% of OA during winter pollution events in Dublin (Lin et al., 2018). However,

the question remains on how these reference profiles vary with stove type and what uncertainties this variation causes in the ME-2 modeling…"

Line 211 why only wood, peat, and smoky coal?

Response: Please also see the reply to the previous comment. Wood, peat, and smoky coal are important OA sources as shown in our previous study (Lin et al., 2017; Lin et al., 2018; Trubetskaya et al., 2021). We have now cited these papers in support of our ME-2 analysis.

Line 216 Could the authors explicitly mention the number of factors identified? What about cooking OA? How the solution with an additional factor looked like? Maybe add details about how the solution was obtained would be added in the supplement.

Response: Please also see the comments above, we have now added more details about how the solution was obtained in the supplement including unconstrained PMF and ME-2 solution with an additional factor. Previous studies conducted at the same sampling site show the aerosol population was mainly affected by the heating emissions but with a relatively minor contribution from traffic or cooking emissions (Lin et al., 2020; Lin et al., 2018).

With an additional factor (i.e., the 6-factor solution), ME-2 separates another OOA factor (i.e., OOA2) but with missing m/z 41 in the mass spectrum which was likely due to the splitting from the already identified OOA factor (Figure R6 or Figure S9 in the revised supplement). The time series of the sum of the 5-factor solution factors was in good agreement with the input OA (Figure R7 or Figure S10 in the revised supplement), suggesting the ME-2 5-factor solution explained the dataset well with no significant residuals.

It now reads, "Increasing the number of factors during the ME-2 analysis (i.e., the 6-factor solution; Fig. S9) identified an additional OOA factor (OOA2) which, however, featured a very low signal at $m/z$ 41 in the normalized mass spectra but a similar signal level at $m/z$ 43 with the already identified OOA factor (Fig. S7). The unambiguous separation of two OOA types requires further research. Nevertheless, for the 5-factor solution, the good correlation ($R^2$=1, slope = 0.99; Fig. S10) between the time series of the explained fraction and the PMF input suggests the 5-factor solution explained the input matrix well."

[Figure]

Figure R6. Mass spectral profile of the 6-factor solution. OOA2 featured a negligible contribution from m/z 41 likely due to the splitting from the already identified OOA.

[Figure]

Figure R7. (a) Time series of the PMF input and the sum of the ME-2 5-factor solution; and (b) Scatter plot between the MF input and the sum of the ME-2 5-factor solution. Also shown in (b) are the linear correlation ($R^2$) and slope.

Line 218 I think the methods section there should definitely be a more detailed description of the city/sampling site. Is it correct to associate HOA to oil heating? Would not be that it is actually from traffic emissions? Section 3.2.2 is about ambient OA, Is Dublin a remote site where there is no important traffic emissions or cooking activities?

Response: Please see the reply to the previous comment, we have provided a more detailed description of the city/sampling site in the revised manuscript.

Dublin is a city in western Europe with a population of around 1 million. Despite Dublin being a large city, the impact from traffic also depends on the distance from the roads, wind speed, wind direction, etc. To evaluate the impact of traffic as well as cooking emissions on urban air quality, our previous study (Lin et al., 2020) simultaneously measured the chemical composition of $PM_1$ at two different sites in Dublin, with one at the roadside in the city center and the other at the same residential site in this study. It was found that, while the diurnal cycle of HOA at the roadside shows typical rush hour peaks, the HOA

at the same urban background shows no clear traffic-related patterns. The latter confirms our conclusion that the traffic emission contribution to HOA at the residential site is minor. The same is true for cooking emissions, with no cooking-like factor showing mealtime spikes found at the residential site.

It now reads, "The ACSM sampling site is ~5 km south to the Dublin city center and is ~500 m away from the nearby road (Fig. S1). ACSM measurements were conducted on the roof of the O'Brien Centre for Science building (~ 30 m above the ground). Previous studies conducted at the same sampling site show the aerosol population was mainly affected by the heating emissions but with a relatively minor contribution from traffic or cooking emissions (Lin et al., 2020; Lin et al., 2018)."

Line 328 If I'm correct, in this study a 5-factor solution is the chosen one, did Lin et al (2018) also selected a 5-factor solution? Are the same 5 factors identified in both studies? I would like to see in the supplement how the 5-factor solution was selected in this study, how the 6-factor solution looks like? Is there another BBOA factor that might hold the OA concentrations that are not attributed to the target profiles used in the constraint?

Response: Yes, a 5-factor solution was also selected in Lin et al., (2018). We have now added more details on how the 5-factor solution was selected in the supplement (please also see the comments to the general comment above). The 6-factor solution separates another OOA factor (i.e., OOA2) but with missing m/z 41 in the mass spectrum which was likely due to the splitting from the already identified OOA factor (Figure R6 or Figure S9 in the revised supplement). There was no other BBOA factor in the 6-factor solution (Figure R6). The time series of the sum of the 5-factor solution factors was in good agreement with the input OA (Figure R7 or Figure S10 in the revised supplement), suggesting the ME-2 5-factor solution explained the dataset well with no significant residuals.

Line 230 There is no description of bootstrap in Method section.

Response: We have now described the bootstrap in the Method section (please also see the reply to the previous comment). In the Method section, it now reads, "Through bootstrapping, a set of new input matrix was created by random resampling of rows from the original ones (Paatero et al., 2014). By randomly duplicating some time points while excluding others, the original dimension of the input matrix was preserved."

253 It is not clear to see the OOA spikes during evening and night time in Figure 3c, maybe show a diurnal plot instead. Moreover, the evening OOA spikes would be related also to boundary layer. It is not clear to me how to relate the evening peaks with the results of using the dilutor.

Response: We have now shown the diurnal plot of OOA and other primary factors (see Figure R5 above or Figure S8 in the supplementary). We agree that a shallower boundary layer can increase the concentrations of OOA in the evening. However, both OOA and POA factors peaked at the same time resulting in a similar diurnal pattern of OOA with the POA factors (Fig. R5), suggesting most OOA in the evening was mostly from heating emissions (condensation of semi-volatile organic compounds or through night-time chemistry (Tiitta et al., 2016)).

Regarding the dilutor, we have now separated the discussion on evening peaks and the use of the dilutor in different paragraphs to reduce confusion.

It now reads, "Moreover, the time series of OOA showed spikes concurrent with primary factors (Fig. S8) during the evening and night-time, suggesting OOA was probably associated with the condensation

of semi-volatile species and/or aging of primary emissions in the real atmosphere"

Line 268 What do the authors mean by "their high emission factors"?

Response: In our previous emission factor study by Trubetskaya et al. (2021), we showed that peat (38-92 g GJ$^{-1}$) and wood (44-179 g GJ$^{-1}$) had higher emission factors than smoky coal (17-29 g GJ$^{-1}$), smokeless coal (5-18 g GJ$^{-1}$), biomass briquettes (7-28 g GJ$^{-1}$). Therefore, despite the small use of peat and wood, their high emission factors make these fuels important factors driving the pollution events.

In the text, it now reads, "Trubetskaya et al. (2021) showed peat (38-92 g GJ$^{-1}$) and wood (44-179 g GJ$^{-1}$) had higher emission factors than smoky coal (17-29 g GJ$^{-1}$), smokeless coal (5-18 g GJ$^{-1}$), biomass briquettes (7-28 g GJ$^{-1}$). Therefore, despite the small use of peat and wood, their high emission factors make these fuels important factors driving the pollution events observed during the heating season."

Line 266 If there were found high concentrations of 300 ug.m-3 and ~50% is from solid fuels, does this mean ~150 ug.m-3 of solid fuel OA is produced from 10% of the population only?

Response: Yes. Our previous study (Lin et al., 2018) provides more details about how the small use of solid fuel caused extreme pollution events.

Line 287 What about m/z 73?

Response: See the comment above, we mentioned m/z 73 can be a marker ion but discussed mostly m/z 60 in the text.

Citation: https://doi.org/10.5194/amt-2021-174-RC1

References:

Canonaco, F., Crippa, M., Slowik, J.G., Baltensperger, U., Prévôt, A.S.H., 2013. SoFi, an IGOR-based interface for the efficient use of the generalized multilinear engine (ME-2) for the source apportionment: ME-2 application to aerosol mass spectrometer data. Atmospheric Measurement Techniques 6, 3649-3661.

CSO, 2016. (Central Statistics Office). Private Households in Permanent Housing Units. https://www.cso.ie/px/pxeirestat/Statire/SelectVarVal/Define.asp?maintable=E4015&PLanguage=0 (accessed on 1 September 2021).

Cubison, M.J., Ortega, A.M., Hayes, P.L., Farmer, D.K., Day, D., Lechner, M.J., Brune, W.H., Apel, E., Diskin, G.S., Fisher, J.A., Fuelberg, H.E., Hecobian, A., Knapp, D.J., Mikoviny, T., Riemer, D., Sachse, G.W., Sessions, W., Weber, R.J., Weinheimer, A.J., Wisthaler, A., Jimenez, J.L., 2011. Effects of aging on organic aerosol from open biomass burning smoke in aircraft and laboratory studies. Atmospheric Chemistry and Physics 11, 12049-12064.

Dall'Osto, M., Ovadnevaite, J., Ceburnis, D., Martin, D., Healy, R.M., O'Connor, I.P., Kourtchev, I., Sodeau, J.R., Wenger, J.C., O'Dowd, C., 2013. Characterization of urban aerosol in Cork city (Ireland) using aerosol mass spectrometry. Atmospheric Chemistry and Physics 13, 4997-5015.

Kourtchev, I., Hellebust, S., Bell, J.M., O'Connor, I.P., Healy, R.M., Allanic, A., Healy, D., Wenger, J.C., Sodeau, J.R., 2011. The use of polar organic compounds to estimate the contribution of domestic solid fuel combustion and biogenic sources to ambient levels of organic carbon and PM2.5 in Cork Harbour, Ireland. Science of the Total Environment 409, 2143-2155.

Lee, T., Sullivan, A.P., Mack, L., Jimenez, J.L., Kreidenweis, S.M., Onasch, T.B., Worsnop, D.R., Malm, W., Wold, C.E., Hao, W.M., Collett, J.L., 2010. Chemical Smoke Marker Emissions During Flaming and Smoldering Phases of Laboratory Open Burning of Wildland Fuels. Aerosol Science and Technology 44, i-v.

Lin, C., Ceburnis, D., Hellebust, S., Buckley, P., Wenger, J., Canonaco, F., Prévôt, A.S.H., Huang, R.-J., O'Dowd, C., Ovadnevaite, J., 2017. Characterization of primary organic aerosol from domestic wood, peat, and coal burning in Ireland. Environmental Science & Technology 51, 10624-10632.

Lin, C., Ceburnis, D., Huang, R.J., Xu, W., Spohn, T., Martin, D., Buckley, P., Wenger, J., Hellebust, S., Rinaldi, M., Facchini, M.C., O'Dowd, C., Ovadnevaite, J., 2019. Wintertime aerosol dominated by solid-fuel-burning emissions across Ireland: insight into the spatial and chemical variation in submicron aerosol. Atmospheric Chemistry and Physics 19, 14091-14106.

Lin, C., Ceburnis, D., Xu, W., Heffernan, E., Hellebust, S., Gallagher, J., Huang, R.J., O'Dowd, C., Ovadnevaite, J., 2020. The impact of traffic on air quality in Ireland: insights from the simultaneous kerbside and suburban monitoring of submicron aerosols. Atmospheric Chemistry and Physics 20, 10513-10529.

Lin, C., Huang, R.-J., Ceburnis, D., Buckley, P., Preissler, J., Wenger, J., Rinaldi, M., Facchini, M.C., O'Dowd, C., Ovadnevaite, J., 2018. Extreme air pollution from residential solid fuel burning. Nature Sustainability 1, 512-517.

NAEI, 2017. (National Atmospheric Emission Inventory). naei.beis.gov.uk (accessed on 1 May 2020).

Ng, N.L., Herndon, S.C., Trimborn, A., Canagaratna, M.R., Croteau, P.L., Onasch, T.B., Sueper, D., Worsnop, D.R., Zhang, Q., Sun, Y.L., Jayne, J.T., 2011. An Aerosol Chemical Speciation Monitor (ACSM) for routine monitoring of the composition and mass concentrations of ambient aerosol. Aerosol Sci. Technol. 45, 780-794.

Paatero, P., Eberly, S., Brown, S.G., Norris, G.A., 2014. Methods for estimating uncertainty in factor analytic solutions. Atmospheric Measurement Techniques 7, 781-797.

SEAI, 2018. (Sustainable Energy Authority of Ireland). Energy Statistics in Ireland. https://www.seai.ie/resources/seai-statistics/key-statistics (accessed on 1 September 2021).

Tiitta, P., Leskinen, A., Hao, L., Yli-Pirilä, P., Kortelainen, M., Grigonyte, J., Tissari, J., Lamberg, H., Hartikainen, A., Kuuspalo, K., Kortelainen, A.M., Virtanen, A., Lehtinen, K.E.J., Komppula, M., Pieber, S., Prévôt, A.S.H., Onasch, T.B., Worsnop, D.R., Czech, H., Zimmermann, R., Jokiniemi, J., Sippula, O., 2016. Transformation of logwood combustion emissions in a smog chamber: formation of secondary organic aerosol and changes in the primary organic aerosol upon daytime and nighttime aging. Atmospheric Chemistry and Physics 16, 13251-13269.

Trubetskaya, A., Lin, C., Ovadnevaite, J., Ceburnis, D., O'Dowd, C., Leahy, J.J., Monaghan, R.F.D., Johnson, R., Layden, P., Smith, W., 2021. Study of Emissions from Domestic Solid-Fuel Stove Combustion in Ireland. Energy & Fuels.

---

## Author Comment (AC2)

We thank the reviewers for their careful examination of our manuscript, and the insightful comments which have helped to improve our manuscript substantially. Below we provided a point-to-point response to the reviewers' comment, where the reviewers' comments are in black, and our responses are in blue.

Reviewer #2

Lin et al. present an interesting analysis of the contributions of solid fuel combustion for heating to PM levels in Dublin. PMF ME-2 modeling is applied to observations from an ACSM using two approaches (conventional "a" value and limits approaches) to assess differences and uncertainties in resulting source apportionments. Both the study findings re: the importance of solid fuel combustion as an evening PM source and more methodological findings re: the ME-2 approaches will be of interest to AMT readers. I have a number of comments for the authors to consider:

Response: We thank the reviewer for the positive comment. See below for a point-to-point response to the comment.

Line 23: It would be better here to refer to normalized peak intensities.

Response: Now referred. It now reads, "…the peak intensities obtained at specific $m/z$ values in the normalized mass spectra were not constant…"

Line 25: It would be better to refer to Positive Matrix Factorization (PMF) analysis using the Multilinear Engine algorithm (ME-2)

Response: Now referred. It now reads, "Using the OA mass spectra of peat, wood, and coal as anchoring profiles and the variation of individual $m/z$ values for the upper/lower limits (the limits approach) in the Positive Matrix Factorization (PMF) analysis with the Multilinear Engine algorithm (ME-2), the respective contributions of these fuels to ambient sub-micron aerosols during a winter period in Dublin were evaluated and compared with the conventional $a$ value approach."

It would be helpful for the authors to provide more detailed information about the fuels burned. For example, there are many types of wood. What wood type(s) are burned in Dublin and what type(s) were used in stove testing? How about peat?

Response: We have now provided details about the fuels burned in the Method Section. It now reads, "Five fuel types were tested including wood, peat, smoky coal, biomass briquettes, and smokeless coal (Table 1). Specifically, wood logs were cut from softwood grown in Ireland; peat was obtained from the peatland in Leitrim, Ireland, and was naturally dried before testing; smoky coal (Silesia, Poland) were purchased from local retail outlets (Trubetskaya et al., 2021); biomass briquettes and smokeless coal (i.e., Ecobrite ovoids) were manufactured at Arigna Fuels (Carrick on Shannon, Ireland)."

Line 87: chloride in biomass burning PM is often present as KCl which vaporizes slowly on the AMS vaporizer, requiring a non-standard treatment of the AMS data. See, for example, the SI in Lee et al. (2010) Aerosol Sci. Tech.; doi: 10.1080/02786826.2010.499884. How was this issue accounted for in ACSM processing for the Dublin study?

Response: In this study, we focused on the mass spectral profiles of organic aerosol, and the issue of the slow vaporization of KCl was not accounted for in ACSM processing. In the method section, we pointed out such concerns, it now reads, "Note that chloride in the aerosol emission from biomass burning is

often present as KCl which vaporizes slowly at 600 °C, requiring a non-standard treatment of the ACSM chloride data (Lee et al., 2010). In this study, we focused on the mass spectral profiles of OA emissions, and the slow vaporization issue was not accounted for (Lee et al., 2010)."

I would like to see a more robust analysis of the comparison between the study site PM1 data and the EPA Ireland PM2.5 site data. Only an R-squared value is given. I suggest adding a scatter plot to SI and discussing not just the correlation but also the slope and intercept of the relationship between the PM1 and PM2.5 measurements. High correlation does not necessarily imply similar PM values at the two sites, just similar temporal variability. Are there other Dublin PM data that could be used to look beyond two locations (e.g., Purple Air monitors) to make a more general assessment of urban PM spatial variability?

Response: We have now added a scatter plot to SI. Slope and intercept of the relationship between $PM_1$ and $PM_{2.5}$ are also provided (See Figure R4 above or Figure S1 in the revised SI). It now reads, "The slope of 0.74 for the linear relationship between $PM_1$ and $PM_{2.5}$ suggests $PM_1$ on average accounts for 74% of $PM_{2.5}$. However, during pollution events, the values of $PM_1$ and $PM_{2.5}$ are very similar, indicating most PM are in the submicron size ranges."

For the current sampling period (i.e., winter of 2016-2017), other Dublin PM data are not available and, thus, we will not extend the discussion. However, during the heating season in 2018 as shown in Lin et al. (2020), $PM_1$ (ACSM+AE33) at a third sampling site (i.e., the roadside in the Dublin city center) also shows an elevated concentration at a similar level as found at the two sampling sites as presented in this study, suggesting heating emissions impact a large area in Dublin.

m/z 60 in the AMS comes from multiple anhydrosugars, including levoglucosan, mannosan, and galactosan – and probably from other similarly structured molecules, too (see, for example, the Lee et al. (2010) paper). This should be stated in the manuscript. Levoglucosan is produced during combustion of cellulose. Mannosan and galactosan come from pyrolysis of hemi-cellulose. The authors should also point, therefore, to the combined fuel content of cellulose and hemi-cellulose, in discussing relationships between fuel and m/z 60.

Response: We thank the reviewer for pointing this out. We have now pointed out the relationship between the fuel content of cellulose/hemicellulose and the observed signal at m/z 60. It now reads, "Mass fragment at m/z 60 (mostly from the $C_2H_4O_2^+$) is due to the fragmentation of anhydrosugars (e.g., levoglucosan, mannosan, and galactosan from the combustion of cellulose/hemicellulose; (Lee et al., 2010)) in the ACSM, and it is, therefore, commonly used as a marker for biomass burning in the AMS/ACSM studies (Cubison et al., 2011; Lee et al., 2010). Therefore, the differences in the content of cellulose/hemicellulose in the test fuels partly contribute to the differences in ion intensity at m/z 60 in ACSM."

Line 173: the text in this line needs to be corrected. I think an extra "m/z" was inadvertently included.

Response: Corrected. "m/z" is now removed.

Line 175: Please define your term **difference ratio** mathematically in the text.

Response: It is now defined, it reads, "…calculated by $(f_{m/z, \text{ stove y}} - f_{m/z, \text{ stove x}}) / f_{m/z, \text{ stove x}}$ where $f_{m/z}$ represents the fraction of the measured m/z to the total organic signal, while stove y represents the Ecodesign or the Boiler stove, and stove x represents the conventional stove; Fig. S3 and S4)…"

2nd half of P. 6: Discussions here about reduced fractional abundance of certain ions are oversimplified. Since f60, for example, refers to the fractional abundance of m/z60, a reduction in f60 is expected when the amount of its solid fuel precursors (cellulose + hemicellulose +?) decreases **more than** the amount of precursors for other m/z ions observed in the AMS. The same is true for the fractional abundance of PAH-related fragments.

Response: We agree that reduced *f60* was likely due to the relatively large reduction of e.g., cellulose/hemicellulose content in the fuel compared to other precursors since the mass spectra are normalized. In the text, it now reads, "The reduced *f60* in the normalized mass spectra for smoky/smokeless coal is likely due to the breakdown of e.g., cellulose during coal formation over millions of years (Höök, 2012), resulting in a relatively low content of cellulose, while accumulating other carbon-rich content, leading to the observed ions at other m/z's. As a comparison, the large contribution from the fragments at *m/z* 77, 91, and 115 suggests a high content of aromatic/polycyclic aromatic hydrocarbon (PAH) compounds in the smoky coal burning emissions."

The diurnal cycle of secondary inorganic species (ammonium, sulfate, nitrate) in the ACSM obs and the source profiles is interesting. Can the authors say more about the origin of these secondary components? Does a combination of ammonia emissions from the solid fuel combustion and lower evening temperatures drive up ammonium nitrate formation in the evening? How much primary sulfate was observed in the stove emissions?

Response: We thank the reviewer for pointing this out. We have now discussed more on the secondary inorganic species (i.e., ammonium, sulfate, nitrate) in the revised manuscript. Specifically, the diurnal pattern of ammonium, nitrate, and sulfate all showed peaks at the same evening hours (20:00-22:00) with the BC (Figure R1 below or Figure S6 in the revised SI), suggesting they may also be related to heating emissions combined with low temperatures. In the stove emission, sulfate was found to contribute to <1% of $PM_1$ for wood burning combustions but the fraction of sulfate was up to 21% of $PM_1$ for smokeless coal, reflecting the higher content of sulfur in the raw fuel (Trubetskaya et al., 2021). However, since ammonia data are not available for both the combustion and ambient studies, we tend not to discuss more on the ammonium nitrate formation, which will be the focus of future studies.

It now reads, "…the diurnal pattern of ammonium, nitrate, and sulfate all showed peaks at the same evening hours with the BC (Fig. 4a), suggesting they may also be related to heating emissions coupled with low temperatures in the evening (Fig. S6). Sulfate was likely associated with the primary emissions from solid fuel combustion (Trubetskaya et al., 2021) given that, in the stove emission, sulfate was found to contribute to <1% of $PM_1$ for wood burning but the fraction of sulfate was up to 21% of $PM_1$ for smokeless coal burning, reflecting the higher content of sulfur in the raw fuel (Trubetskaya et al., 2021). This is consistent with our previous study (Lin et al., 2019), where we demonstrated that sulfate, nitrate, and ammonium can be locally emitted/formed, as well as regionally transported, through the comparison of the ACSM measurement at the same Dublin sampling site and at Carnsore Point, a regional background site…"

[Figure]

Figure R1. Zoomed-in plot of the averaged diurnal cycle of sulfate ($SO_4$), ammonium ($NH_4$), and nitrate ($NO_3$) in µg m$^{-3}$, as well as the temperature ($^o$C) over the entire sampling period.

page 8: In addition to the daily breakdown of OA contributors, please provide the breakdown for the evening period when stove emissions dominate.

Response: The original Fig. 5f and 5g show the relative contribution of the resolved factors during the evening hours (20:00-23:00) when stove emissions dominate. However, it was not clear in the original text. In the revised text, we have now clarified this. It now reads "…both approaches showed peat burning being the largest OA factor (39% (Fig. 5f) vs 41% (Fig. 5g)), followed by HOA (24 vs 25%), OOA (20 vs 18%), wood (14% vs 11%), and coal (4% vs 5%) during the evening hours (20:00-23:00) when stove emissions dominate…"

Lines 272-273: Please add a citation to the statement regarding CO2 capture during biomass growth.

Response: Added. It now reads, "…through photosynthesis during growing as is released when biomass is burned (Marland, 2010)…"

Lines 283-284: I think the authors mean to say "Therefore, extending the ban *to* the use of peat and wood…." As written it sounds as if they are referring to extending an existing ban on peat and wood burning when I think they mean to suggest extending the current ban on burning smoky coal to also forbid peat and wood burning.

Response: Thank you for pointing this out. It is now changed to "…Therefore, extending the ban to the use of peat and wood…"

**Citation**: https://doi.org/10.5194/amt-2021-174-RC2

References:

Canonaco, F., Crippa, M., Slowik, J.G., Baltensperger, U., Prévôt, A.S.H., 2013. SoFi, an IGOR-based interface for the efficient use of the generalized multilinear engine (ME-2) for the source apportionment: ME-2 application to aerosol mass spectrometer data. Atmospheric Measurement Techniques 6, 3649-3661.

CSO, 2016. (Central Statistics Office). Private Households in Permanent Housing Units. https://www.cso.ie/px/pxeirestat/Statire/SelectVarVal/Define.asp?maintable=E4015&PLanguage=

0 (accessed on 1 September 2021).

Cubison, M.J., Ortega, A.M., Hayes, P.L., Farmer, D.K., Day, D., Lechner, M.J., Brune, W.H., Apel, E., Diskin, G.S., Fisher, J.A., Fuelberg, H.E., Hecobian, A., Knapp, D.J., Mikoviny, T., Riemer, D., Sachse, G.W., Sessions, W., Weber, R.J., Weinheimer, A.J., Wisthaler, A., Jimenez, J.L., 2011. Effects of aging on organic aerosol from open biomass burning smoke in aircraft and laboratory studies. Atmospheric Chemistry and Physics 11, 12049-12064.

Dall'Osto, M., Ovadnevaite, J., Ceburnis, D., Martin, D., Healy, R.M., O'Connor, I.P., Kourtchev, I., Sodeau, J.R., Wenger, J.C., O'Dowd, C., 2013. Characterization of urban aerosol in Cork city (Ireland) using aerosol mass spectrometry. Atmospheric Chemistry and Physics 13, 4997-5015.

Höök, M., 2012. Coalcoaland Peatpeat: Global Resourcespealglobal resourcesPeatGlobal Resourcesand Future Supplypealfuture supplyPeatFuture Supply, in: Meyers, R.A. (Ed.), Encyclopedia of Sustainability Science and Technology. Springer New York, New York, NY, pp. 2173-2194.

Kourtchev, I., Hellebust, S., Bell, J.M., O'Connor, I.P., Healy, R.M., Allanic, A., Healy, D., Wenger, J.C., Sodeau, J.R., 2011. The use of polar organic compounds to estimate the contribution of domestic solid fuel combustion and biogenic sources to ambient levels of organic carbon and PM2.5 in Cork Harbour, Ireland. Science of the Total Environment 409, 2143-2155.

Lee, T., Sullivan, A.P., Mack, L., Jimenez, J.L., Kreidenweis, S.M., Onasch, T.B., Worsnop, D.R., Malm, W., Wold, C.E., Hao, W.M., Collett, J.L., 2010. Chemical Smoke Marker Emissions During Flaming and Smoldering Phases of Laboratory Open Burning of Wildland Fuels. Aerosol Science and Technology 44, i-v.

Lin, C., Ceburnis, D., Hellebust, S., Buckley, P., Wenger, J., Canonaco, F., Prévôt, A.S.H., Huang, R.-J., O'Dowd, C., Ovadnevaite, J., 2017. Characterization of primary organic aerosol from domestic wood, peat, and coal burning in Ireland. Environmental Science & Technology 51, 10624-10632.

Lin, C., Ceburnis, D., Huang, R.J., Xu, W., Spohn, T., Martin, D., Buckley, P., Wenger, J., Hellebust, S., Rinaldi, M., Facchini, M.C., O'Dowd, C., Ovadnevaite, J., 2019. Wintertime aerosol dominated by solid-fuel-burning emissions across Ireland: insight into the spatial and chemical variation in submicron aerosol. Atmospheric Chemistry and Physics 19, 14091-14106.

Lin, C., Ceburnis, D., Xu, W., Heffernan, E., Hellebust, S., Gallagher, J., Huang, R.J., O'Dowd, C., Ovadnevaite, J., 2020. The impact of traffic on air quality in Ireland: insights from the simultaneous kerbside and suburban monitoring of submicron aerosols. Atmospheric Chemistry and Physics 20, 10513-10529.

Lin, C., Huang, R.-J., Ceburnis, D., Buckley, P., Preissler, J., Wenger, J., Rinaldi, M., Facchini, M.C., O'Dowd, C., Ovadnevaite, J., 2018. Extreme air pollution from residential solid fuel burning. Nature Sustainability 1, 512-517.

Marland, G., 2010. Accounting for Carbon Dioxide Emissions from Bioenergy Systems. Journal of Industrial Ecology 14, 866-869.

NAEI, 2017. (National Atmospheric Emission Inventory). naei.beis.gov.uk (accessed on 1 May 2020).

Ng, N.L., Herndon, S.C., Trimborn, A., Canagaratna, M.R., Croteau, P.L., Onasch, T.B., Sueper, D., Worsnop, D.R., Zhang, Q., Sun, Y.L., Jayne, J.T., 2011. An Aerosol Chemical Speciation Monitor (ACSM) for routine monitoring of the composition and mass concentrations of ambient aerosol. Aerosol Sci. Technol. 45, 780-794.

Paatero, P., Eberly, S., Brown, S.G., Norris, G.A., 2014. Methods for estimating uncertainty in factor analytic solutions. Atmospheric Measurement Techniques 7, 781-797.

SEAI, 2018. (Sustainable Energy Authority of Ireland). Energy Statistics in Ireland. https://www.seai.ie/resources/seai-statistics/key-statistics (accessed on 1 September 2021).

Tiitta, P., Leskinen, A., Hao, L., Yli-Pirilä, P., Kortelainen, M., Grigonyte, J., Tissari, J., Lamberg, H., Hartikainen, A., Kuuspalo, K., Kortelainen, A.M., Virtanen, A., Lehtinen, K.E.J., Komppula, M., Pieber, S., Prévôt, A.S.H., Onasch, T.B., Worsnop, D.R., Czech, H., Zimmermann, R., Jokiniemi, J., Sippula, O., 2016. Transformation of logwood combustion emissions in a smog chamber: formation of secondary organic aerosol and changes in the primary organic aerosol upon daytime and nighttime aging. Atmospheric Chemistry and Physics 16, 13251-13269.

Trubetskaya, A., Lin, C., Ovadnevaite, J., Ceburnis, D., O'Dowd, C., Leahy, J.J., Monaghan, R.F.D., Johnson, R., Layden, P., Smith, W., 2021. Study of Emissions from Domestic Solid-Fuel Stove Combustion in Ireland. Energy & Fuels.